

# Stress responses upon starvation and exposure to bacteria in the ant *Formica exsecta*

Dimitri Stucki[1,2], Dalial Freitak[1,2,3], Nick Bos[1,2,4] and Liselotte Sundström[1,2]

[1] Organismal and Evolutionary Biology Research Programme/Faculty of Biological and Environmental Sciences, University of Helsinki, Helsinki, Finland
[2] Tvärminne Zoological Station, University of Helsinki, Hanko, Finland
[3] Institute of Biology, Division of Zoology, University of Graz, Graz, Austria
[4] Section for Ecology & Evolution, Department of Biology, University of Copenhagen, Copenhagen, Denmark

## ABSTRACT

Organisms are simultaneously exposed to multiple stresses, which requires regulation of the resistance to each stress. Starvation is one of the most severe stresses organisms encounter, yet nutritional state is also one of the most crucial conditions on which other stress resistances depend. Concomitantly, organisms often deploy lower immune defenses when deprived of resources. This indicates that the investment into starvation resistance and immune defenses is likely to be subject to trade-offs. Here, we investigated the impact of starvation and oral exposure to bacteria on survival and gene expression in the ant *Formica exsecta*. Of the three bacteria used in this study, only *Serratia marcescens* increased the mortality of the ants, whereas exposure to *Escherichia coli* and *Pseudomonas entomophila* alleviated the effects of starvation. Both exposure to bacteria and starvation induced changes in gene expression, but in different directions depending on the species of bacteria used, as well as on the nutritional state of the ants.

## BACKGROUND

Organisms are at all times exposed to a wide variety of stresses, necessitating careful regulation of responses to each stress (*Mooney, Winner & Pell, 1991*; *Broom & Johnson, 1993*). Ultimately this generates trade-offs between different stress responses (*Harshman, Hoffmann & Clark, 1999*; *Hoffmann et al., 2005*; *Marshall & Sinclair, 2010*). As a consequence, natural selection on stress resistance often affects multiple traits (*Chapin, Autumn & Pugnaire, 1993*; *Vinebrooke et al., 2004*), which may lead to different evolutionary trajectories, depending on the stresses an organism encounters. Thus, in order to understand the eco-evolutionary mechanisms of stress resistance, it is important to understand how organisms manage their response to multiple stresses.

One of the most crucial and frequent stresses that organisms encounter is food deprivation. Consequently, organisms have evolved ways to prepare for unfavorable

Corresponding author
Dimitri Stucki,
dimitri.stucki@helsinki.fi

periods with reduced access to resources. For insects, the most common ways of accumulating energy reserves is via the glycogen and triglycerides reserves in the fat body (*Chapman, 2012*), from where they can be mobilized when needed. Glycolipoproteins, such as Arylphorin and Vitellogenin serve to store resources in the hemolymph, but they also transport lipids from the fat body to organs. Therefore, in times of resource limitation, an increased production of such proteins may increase the release of stored resources. For energy management, insects also rely on the insulin pathway, which is involved in the release of stored resources (*Wu & Brown, 2006*). Thus, increased expression of genes associated with the insulin pathway could improve resistance to starvation. However, resistance to starvation can also be gained by lowering metabolic activity to reduce energy consumption (*Rion & Kawecki, 2007*). Furthermore, the regulation of genes involved in adaptation to resource limitation likely depends on further factors, such as the presence of parasites and the potential immune responses.

Given that host immune responses are often costly to maintain under food limitation (*Kraaijeveld, Ferrari & Godfray, 2002*; *McKean et al., 2008*), responses to starvation and immune defenses are likely constrained by trade-offs. Immune defenses not only incur a metabolic cost, but are often also damaging to the host organism, due to immunopathology (*Sadd & Siva-Jothy, 2006*). Therefore, selection is likely to favor reduced expression of immunity in periods of low parasite pressure and/or low food availability. Such stress-dependent immunosuppression could render food-deprived individuals prone to infections by opportunistic pathogens (*Dedet & Pratlong, 2000*), and create a trade-off between immune defenses and starvation resistance. In many insects starvation has been associated with reduced immune responses (*Furlong & Groden, 2003*; *Yang, Ruuhola & Rantala, 2007*), and, conversely, parasite infection has been shown to reduce starvation resistance in the host organism (*Hoang, 2001*). This suggests that the nutritional status influences both the expression of immune genes, and the survival of infections.

Here, we investigated a potential trade-off in starvation resistance and immune defenses against bacteria in a natural population of the ant *Formica exsecta*. We expected to find increased mortality of the ants upon exposure to bacteria, which we assumed to become more pronounced after the ants were starved, as a manifestation of the expected trade-off between starvation resistance and immune defenses. The aim of the study was to investigate how this trade-off is reflected in the expression of selected genes linked to starvation resistance and immune defenses. The genes investigated here included potential starvation resistance genes, as well as a set of immune defense genes. We compared worker ants on their ability to withstand food deprivation, and to resist oral exposure to the entomopathogenic gram-negative bacteria *Serratia marcescens* and *Pseudomonas entomophila*, as well as the response in gene expression upon these stresses. These two entomopathogenic bacteria, *S. marcescens* (DB10) and *P. entomophila* (L48), were originally isolated from *Drosophila melanogaster*, and were shown to infect insects through oral ingestion (*Flyg, Kenne & Boman, 1980*; *Dillon et al., 2005*; *Vodovar et al., 2005*, *2006*). Both bacteria are soil dwelling opportunistic pathogens, with a wide range of potential hosts (*Grimont & Grimont, 1978*; *Vallet-Gely et al., 2010*). In addition, as a

non-pathogenic sham-control we used the benign gut-bacterium *Escherichia coli* (K12), originally isolated from a human stool sample (*Bachmann, 1972*), and frequently used as model organism in insect infection assays (*Bartholomay et al., 2004*; *Altincicek et al., 2011*; *Freitak et al., 2014*; *Mikonranta et al., 2014*).

## MATERIAL AND METHODS

### Animals

For this study we used a population of the ant *F. exsecta* located in southwestern Finland in the municipality of Raseborg, Prästkulla (FI) (59°58′44.6″N 23°20′50.9″E). Our sampling procedures are non-destructive and do not cause harm to the field colonies. No federal laws apply to the collection and scientific study of insects in Finland, and the study-site at Prästkulla is not under protection. The specific population consists of nests with multiple reproductive queens (polygyne), hence the relatedness, and consequently the genetic similarity, among workers is low, yet genetic differences between individual nests is not expected to be high (*Sundström, Seppä & Pamilo, 2005*). In August 2013, we collected adult workers and nest material from inside nine nest mounds, and transferred the workers, and ca. 5dl of nest material immediately into nest boxes (30 × 20 × 15 cm), lined with Fluon® (Whitford, Runcorn, UK) in order to prevent the ants from escaping. The ants were allowed to acclimatize to the laboratory conditions for 24 h at ambient room temperature (23 °C), and were fed daily ad libitum with a standardized diet based on agar, honey and egg (*Bhatkar & Whitcomb, 1970*).

### Experimental design

To start the experiment we transferred 20 monomorphic worker ants from each nest into eight Fluon® coated pots (Ø seven cm, h: five cm) with a plaster bottom (eight pots per nest). To maintain humidity, an open 1.5 ml tube filled with water, and a piece of cotton was placed in each pot, so that the relative humidity was maintained at ca. 75%. The ants were also allowed to acclimatize to the experimental pots for 24 h before the start of the experiment, and were fed with ~100 µl of the standard diet. After 24 h of acclimatization dead ants were replaced from the main colony nest boxes to maintain a starting group size of 20 ants per treatment.

At the start of the experiment the eight pots per colony (72 pots, 20 ants each) were divided into four groups that corresponded to four Exposure treatments (18 pots per Exposure) including a control (factor denoted as Exposure). For these four Exposures we supplemented ~100 µl standard diet either with 50 µl of clear LB-medium (Exposure denoted as control-exposure), or with 50 µl of concentrated bacteria growth culture (Exposures: *S. marcescens*, *E. coli* and *P. entomophila*), leading to a final volume of 150 µl of food, which was provided in a detached cap of a plastic test tube. Each day, a new cap was used and the old food was removed from the pots. In order to provide a standardized amount of live bacteria each day, the growth-cultures were renewed daily by inoculation with 30 µl of overnight culture in 10 ml LB-medium at 37 °C without shaking. We concentrated the bacterial culture by spinning the overnight culture tubes for 3 min at 8,000 rpm, removed all liquid LB-medium and subsequently suspended the

remaining pellet in one ml clear LB-medium, which led to a final concentration of ca. $10^8$ cfu/ml for each Exposure.

The experiment featured two phases. For the first 7 days (first phase), the ants in both pots of each Exposure (18 pots per Exposure) were fed daily ~150 μl diet according to the assigned Exposure. After 7 days, the second phase started (factor denoted as Treatment), during which the ants in one of the two pots of each Exposure (nine pots per Exposure and Treatment) were deprived of food for the next 7 days (denoted as starved), whereas those in the other pot continued to receive food on their assigned Exposure (denoted as continuously exposed). This design allowed us to assess the combined effects of oral bacterial exposure and starvation, as the effect of starvation could have been negated, if oral exposure had followed the starvation step.

Throughout the experiment mortality was recorded daily, and dead ants were removed from the pots. To assess the change in gene expression induced by the treatments, we randomly selected three ants per pot for qPCR analysis on day nine after the beginning of the experiment (i.e., 2 days after the onset of the second phase). Pilot experiments suggested that starvation-induced mortality set in around 2 days after removal of the food, as we observed a 30% drop in survival within the first 2 days, which then petered out during the following days. Thus, we expected effects on gene expression to be pronounced at this time point. Each colony constituted one biological replicate per Exposure/Treatment combination. For each pot, three ants were placed in 300 μl Isol-RNA Lysis Reagent (5 PRIME, Hamburg, DE), and cut into small pieces. The samples were then stored at −80 °C until further processing.

## Gene expression analysis

To extract the RNA we homogenized the thawed samples in 600 μl Isol-RNA Lysis Reagent (5 Prime) with two stainless steel beads using a TissueLyser (Qiagen, Hilden, Germany). We then added 400 μl Isol-RNA Lysis Reagent and 150 μl chloroform (Sigma, St. Louis, MO, USA). After mixing we centrifuged the samples for 10 min at 13,000 rpm at 4 °C, and transferred the upper, transparent phase, containing the RNA, to a new, autoclaved 1.5 ml tube, and supplemented with 500 μl isopropanol (Sigma, min. 99%). After mixing, we let the suspended RNA precipitate over night at −20 °C, and then centrifuged the samples for 30 min at 13,000 rpm at 4 °C to sediment the RNA. After removal of the supernatant, we washed the pellet on ice with 500 μl 80% EtOH (Altia Oyj, Helsinki, Finland), and centrifuged for 10 min at 13,000 rpm at 4 °C. After drying the pellet we dissolved the RNA in autoclaved ddH$_2$O. We measured the concentration and quality of the RNA photospectrometrically with a NanoDrop (Peqlab, Fareham, UK), and eliminated possible gDNA contamination by DNase digest (DNase I, RNase-free; Thermo Scientific, Waltham, MA, USA) before cDNA synthesis with a blend of oligo(dT) and random primers (iScript cDNA Synthesis Kit; Bio-Rad, Hercules, CA, USA). For cDNA synthesis we used one μg RNA for each sample, and afterward diluted the resulting 20 μl cDNA in 80 μl autoclaved ddH$_2$O.

As target genes we chose eight genes representing immune defense mechanisms on a broad scale. We included two storage protein coding genes Vitellogenin 1 (Vg1) and

Arylphorin (Aryl), both of which also have immunological functions (*Amdam et al.,
2004*; *Zhu et al., 2009*), one Insulin Receptor (IR3), two antimicrobial peptides
Lipopolysaccharide-binding protein (LPS-bp) and Hymenoptaecin (Hyme), the two
cascade molecules Pro-Phenoloxidase (PPO) and Toll-receptor (Toll), as well as the
enzyme Lysozyme C (LysC), which has a double role in digestion and immune
defense (*Cançado et al., 2007*; *Kajla et al., 2010*). A list of the primer sequences is
provided in the Table S1.

We designed qRT-PCR primers using the online Primer3 internet-based interface
(http://www.ncbi.nlm.nih.gov/tools/primer-blast/) (*Untergasser et al., 2012*). Primers
were designed by the rules of highest maximum efficiency and sensitivity (Table S1),
to avoid formation of self- and hetero-dimers, hairpins, and self-complementarity.
Gene-specific primers were designed on the basis of the sequences obtained from *F. exsecta*
transcriptome (*Johansson et al., 2013*). Q-RT-PCR was performed on 384-well plates
on a CFX384 Touch™ Real Time PCR Detection System (Bio-Rad, Hercules, CA,
USA) using iQ™ SYBR® Green Supermix (Bio-Rad, Hercules, CA, USA), with an
initiation of 3 min at 95 °C, 40 cycles of 15 s at 95 °C for denaturation, followed by 45 s
at 58 °C for annealing/extension, and a final step for 7 min at 95 °C. All the Q-RT-PCR
assays were run using two technical replicates, which were assessed for consistency
and outliers (one case), and subsequently averaged before normalization. Non-detects
(no amplification signal within 40 qPCR cycles) were set to the maximal cycle number
(i.e., $C_t = 40$), or removed in case the second technical replicate showed amplification.

## Statistical analysis

We tested for the effect of ingestion of bacteria (Exposure), and subsequent starvation
(Treatment) on the survival of the ants by using mixed effects Cox proportional-hazard
regression with Exposure (control-exposure, *E. coli*, *P. entomophila*, *S. marcescens*) as
fixed factor, Treatment (Continuously exposed, Starved) as time-dependent factor.
The time-dependent factor captures the fact that the ants of both Treatments were
essentially treated the same during the first phase of the experiment. Furthermore we
included the interaction between the two factors. To account for differences among the
colonies, we used colony as random intercept. As each colony was exposed to each
Treatment and Exposure only once, there is only one unique combination of colony,
Treatment and Exposure, which corresponds to the pot. As each pot reflected the colony,
specifying pot as a random slope would allow for different reaction norms of the colonies
to each treatment. Unfortunately, random slopes are not yet available in the survival
packages for R. A random intercept term is not appropriate, as it would fix the same slope
for all pots (i.e., colonies), and further reduce the power of the analysis, potentially
leading to the acceptance of a false null hypothesis. We therefore decided to omit pot as a
random factor. In case of a significant interaction between Treatment and Exposure,
we examined pairwise mortality risks as post hoc analysis, correcting for multiple
comparisons using false discovery rates (*Benjamini & Hochberg, 1995*). Due to the
limited sample size we focused the statistical inference on a set of planned comparisons:
(i) between starved and continuously exposed individuals within each Exposure and

(ii) between each Exposure and the control exposure within each Treatment. We used a single model including all Exposures to allow the reader the comparison among all Exposures, just without the aid of statistical inference. We tested for overall effects of Treatment, Exposure and the interaction through likelihood ratio tests. We compared the full model against an intercept only model, and against a model with dropped interaction term.

Gene expression was analyzed on the level of normalized $C_t$ values. Three ants were pooled from each pot, which led to a final sample size of nine samples per Treatment and Exposure. To test for differences in gene expression across Exposures and Treatments, we first performed Principal Component Analysis (PCA) on the inverted normalized $C_t$-values. We inverted the $C_t$-values because they are negatively correlated with the transcript level (i.e., higher $C_t$-values indicate lower gene expression levels). Normalization of the $C_t$-values was done with the NORMA-gene algorithm, which does not require reference genes (*Heckmann et al., 2011*). We assessed the data for suitability for a PCA: The Kaiser-Meyer-Olkin measures of the eight genes ranged between 0.61 and 0.84, and thus indicated a sampling adequacy acceptable for PCA (*Field, Miles & Field, 2012*). A Bartlett's test of sphericity also indicated sufficiently strong correlations between the genes ($X^2 = 321.4$, $p < 0.0001$), and the determinant of the correlation matrix (det = 0.009) indicated that the correlations were not too strong for a PCA (i.e., det > 0.00001; *Field, Miles & Field (2012)*).

We used an unrotated PCA for component selection, and consulted the scree plot of the Eigen values for the relative weights of the components. Two components had an Eigenvalue >1.0 (Table S2), but the scree plot showed a high relative weight only for PC1, whereas the weight of PC2 was not distinctively higher compared to the subsequent components (Fig. S1). A preliminary analysis including PC1 and PC2 showed no significant effects of PC2 for any of the factors (Tables S3 and S4). We therefore decided to use only PC1 for the analysis. The PC scores were then used as dependent variables in a linear mixed effects model to test for an association of PC1 with Exposure and Treatment. Colony was added as a random intercept. In case of a significant interaction of Treatment and Exposure, we examined pairwise gene expression differences as post hoc analysis, correcting for multiple comparisons using false discovery rates. We used the same procedure for post hoc comparisons as for the survival data. To assess the contribution of each gene to the effects found for PC1, we here arbitrarily define the loadings as strong (loading 0.67–1.0), moderate (loading 0.33–0.66), or weak (0.0–0.32). Because the PCA indicated multicollinearity among the genes, we did not use a gene-by-gene analysis as a basis for our conclusions, but provide one as Supplementary Material (Fig. S2; Table S5).

We assessed the validity for a cox-proportional hazard model with a proportional hazard test (*Grambsch & Therneau, 1994*), which did not reveal any indication for non-proportional hazards (Chi-squared: 9.17, $p = 0.24$). As a method to test for the presence of proportional hazards is not available for mixed effects Cox-proportional hazard models, we specified a default cox-ph model, with Colony specified as frailty term (*Therneau, Grambsch & Pankratz, 2003*). The validity of the linear mixed effects models

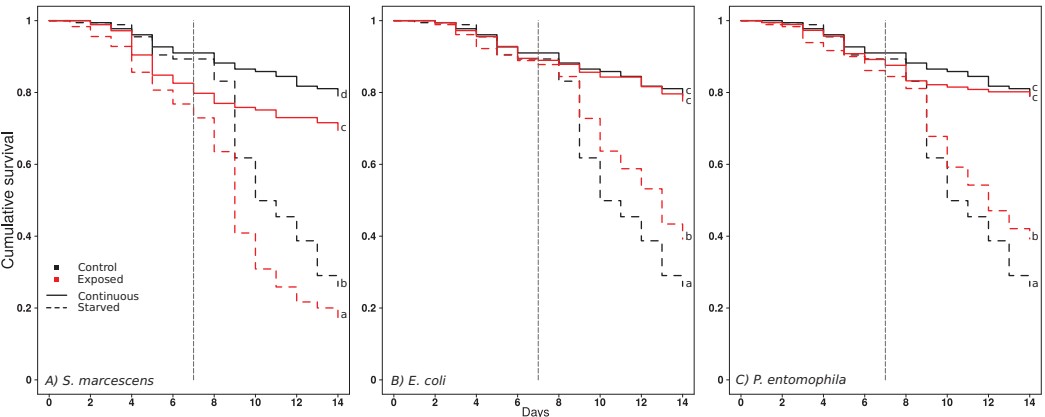

**Figure 1 Effect of treatments on ant surviva.** (A) *S. marcescens*, (B) *E. Coli*, (C) *P. entomophila*. Cumulative survival after oral exposure to bacteria and upon starvation. For each bacteria Exposure, the survival of the ants of the control Exposure is repeatedly shown in black for clarity, and the survival of the bacteria Exposure in red. The ants were either fed daily throughout the experiment (solid lines), or for a limited time followed by starvation (dashed lines). The vertical dashed line indicates the onset of starvation for the starvation treatment. Letters indicate significant differences within each Exposure and are not directly comparable between exposures.

was assessed visually through inspection of the residuals and the leverage, which showed no evidence for misspecification of the model. For all statistical tests we used a threshold level of significance of $\alpha = 0.05$. Statistical analysis was carried out in R version 3.0.2 (*R Core Team, 2015*), using the packages survival (*Therneau, 2013*), coxme (*Therneau, 2018*), psych (*Revelle, 2016*), lme4 (*Bates et al., 2015*), lmerTest (*Kuznetsova, Brockhoff & Christensen, 2017*), and multcomp (*Hothorn, Bretz & Westfall, 2008*).

## RESULTS

### Survival

Overall the Cox-proportional hazard regression revealed significant effects of the treatments on the survival of the ants (Likelihood ratio test: Chi-square = 476.89, d$f$ = 7, $p < 0.0001$). The interaction was close to significant (Likelihood ratio test: Chi-square = 7.41, d$f$ = 3, $p = 0.06$). Given the borderline effect, we decided to retain the interaction term in the model as the log likelihood was slightly higher when the interaction term was included (loglik$_{Full}$ = −4032.0, loglik$_{Reduced}$ = −4035.7).

Starvation reduced the survival of the ants in all exposure regimes (Fig. 1; Tables 1 and 2). Exposure to *S. marcescens* significantly reduced survival compared to the respective controls, both under continuous feeding and starvation (Fig. 1A, comparisons between black and red continuous lines, and dotted lines, respectively; Tables 1 and 2). No significant changes in mortality emerged when the ants were continuously fed and exposed to *E. coli* or *P. entomophila* (Figs. 1B and 1C, comparison between black and red continuous lines; Tables 1 and 2). However, when these ants were starved, they survived significantly longer if they had been exposed to *E. coli* or *P. entomophila*, than if they had been exposed to the control-exposure (Figs. 1B and 1C, comparison between black and red hatched lines; Tables 1 and 2).

**Table 1 Effect of treatments on ant survival.**

|  | β ± SE | z | p |
|---|---|---|---|
| Starved | 2.46 ± 0.20 | 12.26 | <0.0001 |
| *E. coli* | 0.11 ± 0.19 | 0.57 | 0.57 |
| *S. marcescens* | 0.73 ± 0.17 | 4.31 | <0.0001 |
| *P. entomophila* | 0.21 ± 0.18 | 1.16 | 0.25 |
| Starved × *E. coli* | −0.51 ± 0.24 | −2.11 | 0.035 |
| Starved × *S. marcescens* | −0.36 ± 0.23 | −1.58 | 0.11 |
| Starved × *P. entomophila* | −0.61 ± 0.24 | −2.51 | 0.012 |

Note:
Results from the ad hoc Cox proportional-hazard regression on the influence of each factor (and interactions) on the survival of ants. The hazard rate coefficient (β) indicates the change in the probability to die from the treatment/factor compared to the corresponding control treatment/factor. Starvation indicates the comparison of starved to continuously exposed conditions, and the bacteria names correspond to the effect of supplementing the food with *S. marcescens*, *P. entomophila* or *E. coli*, as compared to the control conditions.

**Table 2 Pairwise comparison of ant survival.**

| Exposure/Treatment | β ± SE | z | p |
|---|---|---|---|
| Control/continuous vs control/starved | −2.46 ± 0.20 | −12.26 | <0.0001 |
| *S. marcescens*/continuous vs *S. marcescens*/starved | −1.95 ± 0.20 | −9.58 | <0.0001 |
| *E. coli*/continuous vs *E. coli*/starved | −2.10 ± 0.18 | −11.47 | <0.0001 |
| *P. entomophila*/continuous vs *P. entomophila*/starved | −1.85 ± 0.20 | −9.17 | <0.0001 |
| *S. marcescens*/continuous vs control/continuous | −0.73 ± 0.17 | −4.31 | <0.0001 |
| *S. marcescens*/starved vs control/starved | −0.37 ± 0.15 | −2.50 | 0.0168 |
| *E. coli*/continuous vs control/continuous | −0.11 ± 0.19 | −0.58 | 0.60 |
| *E. coli*/starved vs control/starved | 0.41 ± 0.16 | 2.62 | 0.0141 |
| *P. entomophila*/continuous vs control/continuous | −0.21 ± 0.18 | −1.16 | 0.31 |
| *P. entomophila*/starved vs control/starved | 0.39 ± 0.16 | 2.50 | 0.0168 |

Note:
Results from the planned post hoc pairwise comparisons of survival between Treatments (Continuously exposed, Starved) within each Exposure regime, as well as between each bacterial Exposure regime (Control exposure, *S. marcescens*, *E. coli*, *P. entomophila*) and the control exposure within each Treatment. All *p*-values were adjusted for multiple comparisons using false discovery rates.

## Gene expression

From the PCA on gene expression we retained one component, which explained 51% of the variation (Table S2). The loadings of all genes were positive, and either moderate (Vg1: 0.60; PPO: 0.66; Hyme: 0.36; LPS-bp: 0.62), or strong (Aryl: 0.76; IR3: 0.88; LysC: 0.82; Toll: 0.89). This shows that all genes were regulated in the same direction and to a similar degree (apart from Hymenoptaecin), and that the PC1 scores therefore reflect the general level of gene expression. Overall the mixed effects model revealed significant effects of the treatments on gene expression (Likelihood ratio test: Chi-square = 177.2, d$f$ = 2, $p$ < 0.0001), with a significant interaction term (Likelihood ratio test: Chi-square = 130.81, d$f$ = 3, $p$ < 0.0001). Continuous exposure to *S. marcescens* led to a decrease in gene expression levels, compared to the control-exposure, whereas the opposite occurred when the ants were starved (Fig. 2, comparison between black symbols, and white symbols, respectively; Tables 3 and 4). By contrast, ants that were continuously exposed

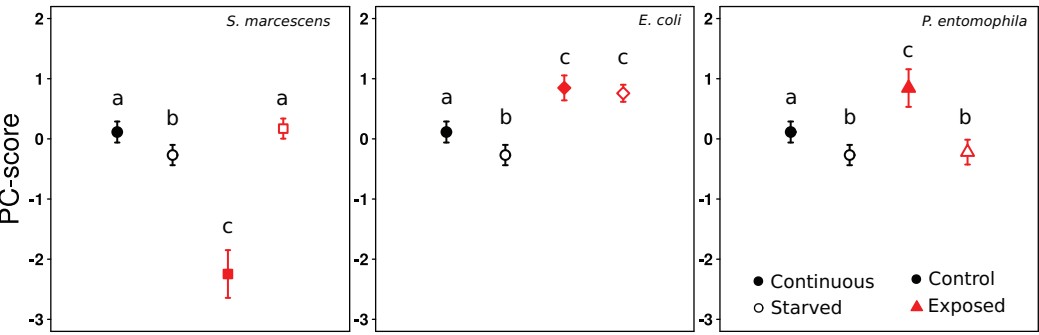

**Figure 2 Effect of treatments on gene expression (PC1).** PC1-scores averaged across colonies, reflecting the expression levels of the eight genes, after 9 days of continuous feeding (filled symbols) or 7 days continuous feeding and 2 days of starvation (open symbols). The food was either supplemented with LB medium (circles), *S. marcescens* (squares), *E. coli* (diamonds), or *P. entomophila* (triangles). The results for the control Exposure are repeatedly shown for each bacteria Exposure for clarity. Error bars show 95% confidence intervals for the mean PC1-score across colonies. Letters indicate significant differences within each Exposure and are not directly comparable between exposures.

**Table 3 Effect of treatments on gene expression.**

|  | β ± SE | t | p |
|---|---|---|---|
| Starved | −0.38 ± 0.15 | −2.64 | 0.0104 |
| *E. coli* | 0.73 ± 0.15 | 5.06 | <0.0001 |
| *S. marcescens* | −2.36 ± 0.15 | −16.27 | <0.0001 |
| *P. entomophila* | 0.73 ± 0.15 | 5.04 | <0.0001 |
| Starved × *E. coli* | 0.29 ± 0.21 | 1.43 | 0.16 |
| Starved × *S. marcescens* | 2.80 ± 0.21 | 13.65 | <0.0001 |
| Starved × *P. entomophila* | −0.68 ± 0.21 | −3.33 | 0.0015 |

**Note:**
Results from an ad hoc linear mixed effects model on the effect of Exposure (Control, *S. marcescens*, *E. coli*, *P. entomophila*) and Treatment (Continuously exposed, Starved) on the scores of the selected Principal Component (PC1) reflecting gene expression. Parameter estimates are given as β plus/minus standard error (SE).

**Table 4 Pairwise comparison of gene expression.**

| Exposure/Treatment | β ± SE | z | p |
|---|---|---|---|
| Control/continuous vs control/starved | −0.38 ± 0.15 | 2.64 | 0.0102 |
| *S. marcescens*/continuous vs *S. marcescens*/starved | 2.42 ± 0.15 | −16.66 | <0.0001 |
| *E. coli*/continuous vs *E. coli*/starved | −0.09 ± 0.15 | 0.62 | 0.61 |
| *P. entomophila*/continuous vs *P. entomophila*/starved | −1.07 ± 0.15 | 7.34 | <0.0001 |
| *S. marcescens*/continuous vs control/continuous | −2.36 ± 0.15 | 16.27 | <0.0001 |
| *S. marcescens*/starved vs control/starved | 0.44 ± 0.15 | −3.03 | 0.0036 |
| *E. coli*/continuous vs control/continuous | 0.73 ± 0.15 | −5.06 | <0.0001 |
| *E. coli*/starved vs control/starved | 1.03 ± 0.15 | −7.08 | <0.0001 |
| *P. entomophila*/continuous vs control/continuous | 0.73 ± 0.15 | −5.04 | <0.0001 |
| *P. entomophila*/starved vs control/starved | 0.05 ± 0.15 | −0.34 | 0.79 |

**Note:**
Results from the planned post hoc pairwise comparisons of gene expression reflected as the scores from the selected component (PC1) of a PCA. Comparisons were performed between Treatments (Continuously exposed, Starved) within each Exposure regime, as well as between each bacterial Exposure regime (*S. marcescens*, *E. coli*, *P. entomophila*) and the control exposure within each Treatment. All p-values were adjusted for multiple comparisons using false discovery rates.

to *E. coli* or *P. entomophila* showed higher gene expression levels than ants of the control-exposure (Fig. 2, comparison between black symbols). When the ants had been exposed to *E. coli*, this also occurred after starvation, but not when they had been exposed to *P. entomophila* (Fig. 2, comparison between white symbols; Tables 3 and 4).

Ants that had received either the control-exposure or *P. entomophila* treatment showed lower gene expression levels after starvation, than during continuous exposure (Fig. 2, comparison between black and white symbols within regime). By contrast, the ants that had been exposed to *S. marcescens*, showed significantly higher gene expression levels in starved, than in continuously exposed individuals. Only ants that had been exposed to *E. coli* showed no difference in gene expression between starved, and continuously exposed individuals (Fig. 2; Tables 3 and 4).

## DISCUSSION

In this study we investigated the responses of the ant *F. exsecta* to two stresses; oral exposure to bacteria and subsequent starvation. As expected, starvation induced a strong decrease in survival, but the responses in combination with oral exposure to bacteria differed depending on the bacteria. We also found that both starvation and exposure influenced gene expression differently depending on the bacteria used for exposure, and the feeding regime. However, we found no evidence for a gene-level trade-off between investment in immune defenses and starvation resistance, as none of the genes showed differential expression patterns (based on the fact that all genes loaded positively on PC1) which indicated that the regulation of gene expression was highly consistent upon each treatment.

As expected both starvation and exposure to *S. marcescens* induced increased mortality under both feeding regimes (continuous food vs starvation during the second stage). As these results were expected and the statistical support is strong, we believe this result to be unaffected by the omission of pot as a random effect. We found no evidence for additive effects from exposure combined with starvation. In contrast, neither *E. coli* nor *P. entomophila* induced severe mortality in the ants. This may indicate that these bacteria are not infectious to *F. exsecta*, either due to a lack of the necessary infective mechanisms, or due to an efficient immune response from the ants. Yet, surprisingly, exposure to *P. entomophila* and *E. coli* alleviated the effects of starvation on mortality, at least when administered orally. This stands in contrast to previous findings in insects, which showed an increase in the susceptibility to infections under food-deprived conditions (*Feder et al., 1997*; *Vass & Nappi, 1998*; *Kraaijeveld, Ferrari & Godfray, 2002*; *McKean et al., 2008*), or a decrease in starvation resistance when infected (*Hoang, 2001*). Because the *p*-values of these results are relatively close to our alpha level of 0.05 it is possible that this result may be biased due to the omission of pot as a random effect. Nevertheless, the fact thet this effect was observed independently for two different bacteria, and that the parameter estimate for *S. marcescens* points in the same direction, suggests that this is valid observation.

A possible explanation for the reduced mortality during the starvation phase could be the induction of anorexia (*Adamo, 2005*; *Adamo, Fidler & Forestell, 2007*; *Ayres &*

*Schneider, 2009*), which in insects can either reduce or increase survival, depending on the pathogen (*Ayres & Schneider, 2009*). In such a scenario, infection leads to loss of appetite, and the ensuing dietary restriction alters metabolic rates, which may increase starvation resistance (*Chippindale et al., 1993*; *Burger et al., 2007*). Similarly, the ants may have sensed the food contamination and reduced consumption, which may have primed their physiological resistance to starvation. Reduced food consumption may also have led to the down-regulation of gene expression in ants that were continuously exposed to *S. marcescens.* Thus, the reduced survival during continuous exposure to *S. marcescens* may also reflect mortality from starvation. However, given that the two other bacteria did not elicit such a response, the ants may not recognize all bacteria or the response to contamination may depend on the bacteria. Finally, both bacteria could have added nutritional value to the diet, which could alleviate the effects of starvation. At present these explanations remain open to debate.

Starvation induced clear effects on gene expression in all treatments, but the patterns of change differed considerably between exposure regimes. Gene expression increased significantly in starved ants, compared to the control, when the ants had been exposed to *E. coli* or *S. marcescens* before starvation, but not in those exposed to *P. entomophila*. Also in non-starved ants the responses differed between the infection regimes, such that gene expression was up regulated, compared to the control-exposure, in ants exposed to *P. entomophila*, and *E. coli*, but strongly down-regulated in those exposed to *S. marcescens*. The responses to *P. entomophila*, and *E. coli* under non-starved conditions probably reflect a general immune response. The high dose may have elicited a general immune response. Alternatively, this may be a prophylactic response upon the detection of contaminated food. The response to *S. marcescens* may reflect a trade-off between energy conservation and immune defense, or reflects an inability to uphold immune defenses under severe infections. In the control-exposure, gene expression decreased upon starvation, which likely reflects a mechanism to save resources under food limitation (*Rion & Kawecki, 2007*). It is important to note, however, that the set of genes investigated here only represent a subset of the genes involved in metabolic regulation and immune defenses. Therefore, other genes involved in these regulation pathways may also have influenced the effects of starvation. We discuss these results, and the comparisons within each exposure regime in detail below.

When comparing starved and non-starved ants exposed to *P. entomophila* we found significantly lower levels of gene expression in the starved group. This could indicate purging of infection, as the ants at this point ceased to receive the supplemented food, and an activation of immune responses therefore was not needed any more. This may also explain the enhanced survival of the exposed ants, compared to the control, if the bacteria conveyed nutritional value. However, a possible effect of additive nutritional value is speculative, and it remains to be tested whether these bacteria provide a relevant quantity of nutritional value to insects. In the ants exposed to *E. coli* an elevated level of gene expression was maintained also under starvation. This may indicate that the naturally occurring *E. coli* was not directly targeted, but that the immune response was maintained due to a higher sensitivity after exposure (*Sadd & Schmid-Hempel, 2006*), or the presence of other pathogens. In this case the enhanced survival of exposed starved

ants, compared to the non-starved ones, remains unexplained, unless the bacteria conveyed nutritional value also in this case. Unlike *E. coli* and *P. entomophila*, *S. marcescens* induced a down-regulation of gene expression in non-starved ants, and an up-regulation in starved ones. As the supply of *S. marcescens* also ceased upon the onset of starvation, the ants may have restored gene expression levels despite starvation, and cleared the infection. This could explain the absence of an additive negative effect of both challenges, infection and starvation.

In addition to behavioral and physiological adaptations against infection and starvation, social insects have access to collective defenses (*Cremer, Armitage & Schmid-Hempel, 2007*), which might explain the absence of a trade-off between survival and immune defenses, both at the level of individuals, and terms of gene expression. If the ants intensified their social defenses (e.g., trophallactic prophylaxis (*Hamilton, Lejeune & Rosengaus, 2011*)) when starved, the individual defenses would have been less strained, and thus a trade-off between starvation resistance and immune defenses would be less evident from the physiological response. Indeed, social interactions were reported to increase when ants were starved for a longer period (*Rueppell & Kirkman, 2005*). If such interactions encompass social immune defenses, such as trophallactic prophylaxis (*Hamilton, Lejeune & Rosengaus, 2011*), this would indicate a synergistic interplay of the defenses against starvation and infections.

The pattern of gene expression was very similar across all investigated genes and treatments (Fig. S2; Table S5). Furthermore, the observed patterns of gene expression were consistent among the colonies, which suggests that these stresses trigger conserved responses that are common to the workers of *F. exsecta*. Given that the candidate genes represented a broad selection of immune defense pathways, this suggests a systemic response to the stresses, and shows the synergistic relationship between different immune pathways, such as the Toll and Imd pathways (*Tanji et al., 2007*). It is, however, possible that the severity of the stresses prevented a more nuanced response among genes, and instead induced a systemic response.

## CONCLUSIONS

In conclusion our results indicate that exposure to some bacteria may have a beneficial effect on starvation resistance in the ant *F. exsecta*. Although the underlying mechanism is not evident from the regulation of the investigated genes, our results highlight the fact that the interplay between nutritional state, and pathogen exposure is not regulated by straightforward trade-offs. Instead the beneficial effect of exposure to bacteria on starvation resistance may reflect the ubiquity of pathogens and resource scarcity in nature. Given that the responses to these stresses were similar across all genes, as well as between colonies, this suggests that evolutionary trajectories due to the presence of multiple stresses can result in conserved stress responses.

## ACKNOWLEDGEMENTS

We thank our field assistants for their valuable help during sampling of the colonies, as well as our suppliers of the bacteria strains, Lauri Mikonranta (University of Jyväskylä;

*S. marcescens* & *E. coli*), and Bruno Lemaitre (University of Lausanne; *P. entomophila*). Furthermore we want to thank the editor Prof. Dr. Joseph Gillespie and two anonymous reviewers for valuable input on the manuscript.

### Funding
This study was funded by the Academy of Finland (grants #252411, #284666) to the Centre of Excellence in Biological Interactions, grants #251337 (to Liselotte Sundström), #289731 (to Nick Bos)), as well as the University of Helsinki. The funders had no role in study design, data collection and analysis, decision to publish, or preparation of the manuscript.

### Grant Disclosures
The following grant information was disclosed by the authors:
Academy of Finland: #252411, #284666.
Centre of Excellence in Biological Interactions: #251337 (to Liselotte Sundström), #289731 (to Nick Bos).
University of Helsinki.

### Competing Interests
The authors declare that they have no competing interests.

### Author Contributions
- Dimitri Stucki conceived and designed the experiments, performed the experiments, analyzed the data, prepared figures and/or tables, authored or reviewed drafts of the paper, approved the final draft.
- Dalial Freitak conceived and designed the experiments, authored or reviewed drafts of the paper, approved the final draft.
- Nick Bos conceived and designed the experiments, authored or reviewed drafts of the paper, approved the final draft.
- Liselotte Sundström conceived and designed the experiments, authored or reviewed drafts of the paper, approved the final draft.

### Data Availability
Raw data are available in the Supplemental Files.

### Supplemental Information
Supplemental information for this article can be found online at http://dx.doi.org/10.7717/peerj.6428#supplemental-information.

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
