# Peer review of "Stress responses upon starvation and exposure to bacteria in the ant Formica exsecta"

_PeerJ, doi:10.7717/peerj.6428_

## Round 0.1 · original submission · Major Revisions

Dear Dr. Stucki and colleagues:

Thanks for submitting your manuscript to PeerJ. I have now received two independent reviews of your work, and as you will see, the reviewers raised some concerns about the research. Despite this, these reviewers are optimistic about your work and the potential impact it will have on the ant research community. Thus, I encourage you to revise your manuscript accordingly, taking into account all of the concerns raised by both reviewers.

I look forward to seeing your revision, and thanks again for submitting your work to PeerJ.

Good luck with your revision,

-joe

Reviewer 1 ·

Basic reporting

Line 44: It might read better if written "serve to store resources in the..."

Line 57: "Incur a metabolic..." may sound better than "cause".

Line 58: I think this is rather immunopathology than autoimmunity.

Line 65-68: I had to read these two sentences several times to understand them fully. I think that they could be written more simply, i.e. that the ability of an organism to survive an infection will likely depend on its nutritional state.

Line 77: ", which are all potential..." might read better.

Line 78: Here, I would suggest removing the names of the genes and just say you look at a subset of immune genes, then in the methods state the gene names along with their function. When I read the ms I was missing an explanation here of the gene's function, which then came later in the methods.

Line 82: "These" instead of "the".

Line 88: Were was this bacteria isolated/obtained from for the study?

Line 136: I'd move up the explanation for choosing this time point to here.

Line 154: Missing a "the" between measured and concentration.

Line 226: Adding headings for the results may be benefical.

Line 262: Can remove the comma after 'bacteria'.

Line 343: Maybe change 'can' to 'may' as the results are 100% conclusive.

Supplemental figure 1: I think you can add a more informative Y-axis label.

Supplemental data sets: I'm not 100% sure how this is handled by PeerJ, but I think you should have a 'legend' for these in the PDF that explains what the column names mean and they type of data each column contains. For example, it's not obvious that Cens is survival and that 0 = alive or 1 = dead. It might also be handy to include the PCA results as a third data set since most of the statistics are carried out directly on these and it's not necessarily straightforward for a third party to run the PCAs again.

Experimental design

104: Honey has antimicrobial properties and so I am surprised that ants were both fed honey containing food, as well as mixing the bacteria with it. Is it possible that the honey neutralised some/all of the bacteria and whether this could explain the absence of an effect of two of the bacteria?

Line 107: I am unsure how polymorphic size is in this species, but I know it can vary greatly within some Formica. Did you control for ant size, given that this may affect the relative dose of bacteria they receive?

Line 107: Throughout, I think it would be helpful to have in brackets the final sample sizes used, as it was hard to follow as is.

Line 117: Please clarify whether this means 50 ul was added to 100 ul, or 50 ul + 50 ul. How was the food provided, e.g. in a dish? I ask because it's a known problem that the bacterial spores can sink to the bottom of larger droplets, possibly leading to less being consumed/lack of infections occurring.

Line 119: Was the old food removed before the new added? Moreover, as far as I can tell, ants were allowed to consume the food freely, rather than administering a controlled dose to each. Do you have any data on how much of the food was consumed on a daily basis and whether enough food was provided per ant (I think the amount you gives works out at about 7 ul of food per ant)? More information is needed here, especially as its known animals can avoid contaminated food.

Line 124: Was this concentration the same across the bacterial spp?

Line 126: I find it peculiar that in one group the ants are continually fed bacteria for an additional seven days. A more balanced design may have been to cease feeding bacteria in the group that continues to have food, otherwise, this group receives a higher bacterial dose than the starved comparison, which may have affected immune gene expression etc.

Line 187: It might be helpful to explain what a time-dependent factor is and why you included it this way.

Line 188: I may be wrong, but I think you also need a random intercept effect for pot since ants in the same pot are non-independent.

Line 193: It's unclear what is meant by nests as a biological replicate, please add an explanation as well as final sample sizes.

Line 212: Unless the ants were pooled (this was unclear), pot probably should be included as a random intercept effect here as well.

Line 216: Is this arbitrary scale approach have any basis in the literature?

Line 218: Was multicollinearity present between all genes or subsets?

Line 224: Please state what model assumptions were checked for the Cox and LMERs and how you assessed model stability. There are some survival curves crossing that may violate the assumption of proportional hazards.

Validity of the findings

Line 227 & Table 1: You are missing an overall effect of bacterial treatment, feeding regime and their interaction. Table 1 appears to be the summary output of the cox regression when the reference category is set to the control, which is good to show for the hazard ratios, but you need to also report the likelihood ratio or Wald statistic of the whole model - this tells us about how well the model fits the data (at the bottom of the summary output) - as well as the statistic and associated p-value for the predictors in the model. This can be done using likelihood ratio tests that compare full and reduced models with terms removed. Currently, it's unclear whether there is a significant interaction present - if not, it should be removed from the model. However, if there is a significant interaction, it's inadvisable to interpret the main effects alone. I believe this section of the results, therefore, needs to be fully revised.

Table 2: It's a bit unclear how these pairwise post hoc comparisons were carried out. Did you use the mult.comp R package where you set up your own contrasts or did you re-level the model with different reference groups until all pairwise combinations were produced? Either is fine, but the results will be slightly different. It might also be good to add to the legend that the p-values were adjusted for multiple comparisons for clarity. It might be worth stating somewhere that you used planned comparisons as you were not interested in comparing between bacteria. This makes me wonder, based on the way the post hocs presented and the graphs, whether it may be simpler to have three separate Cox regressions for the three bacteria? It may be necessary to do this in fact, depending on what the main effects for treatment and feeding are.

Figure 1: I think you should state in the legend that the control groups are the same data in each of the graphs, but that you've split up the data based on bacteria to aid visual interpretation. Could you also add letters to the ends of the lines that indicate significantly different groups? This would make the graph easier to interpret on its own.

Line 231: You could add 'continuously fed and exposed to bacteria' since it's the feeding (or lack of it) that seems to be the important factor.

Line 238: I found this section particularly hard to follow, as it's unclear what the loadings mean. For example, you state that 'all genes responded the same way' but then go on to discuss the differences between them.

Figure 2: Could you change the order of the bacteria so that they are presented in the same way Figure 1. In fact, it may be better to present this data as three separate graphs again, to make it easier to compare with Figure 1. This would also better reflect the planned post hocs in Table 4, which do not compare between bacteria. Currently, Figure 2 invites readers to compare the dots and the letters are not actually representative of a full pairwise comparison set up, so it's misleading to show them all in one. Finally, I'm a bit unsure what the CIs are if the dots are not means/medians? Please explain this more fully so that they can be meainfully interpreted.

Table 1: Again, this seems to be the summary output of the LMER. You need to show a p-value that tests the fit of the model to the data (i.e. by comparing the full model to an intercept-only null model) as well as comparisons to reduced models to give you the results of the main effects (i.e. bacteria exposure, feeding regime and their interaction). Please see this handy tutorial for more info: http://www.bodowinter.com/tutorial/bw_LME_tutorial2.pdf

Line 267: This is a bit unclear, as the pattern does not seem to be the same across treatments, e.g. gene expression is decreased in the SM group?

Line 272: Not sure you can use the word infected here, since you find no mortality effect and could not confirm infection actually took place. 'Exposed' may be better.

Line 275: Possible explanations for any lack of mortality are that the methods you used did not result in any infection. This could be because the bacteria sunk to the bottom of the food, they are simply non-infectious to the ants, they were neutralised in the crop by the ants' formic acid, the ants avoided the bacteria (hence, data on how much ants actually consumed would be handy) etc. It would be good if you could discuss some of these other possibilities.

Line 284: Is there any evidence of bacteria being used as food in insects? I wonder how you could test this in your system.

Line 325: Unclear how cuticle grooming and other external social defences would affect oral infections?

Additional comments

I'm not really sure if this is needed, by I wondered why the immune and metabolic genes were analysed as one? Would it perhaps be informative to runs PCAs on these two sets of genes separately to tease apart any trade-offs between them?

Reviewer 2 ·

Basic reporting

The manuscript is well written, and I found it to be an interesting read. I have only minor points to the basic reporting
Line 110: I assume that the standard diet mentioned in Line 110 is without the addition of agar?
Line 288-289: from figure 1 it appears to me that gene expression in starved ants exposed to E. coli and S. marcescens increased significantly compared to the control and not P. entomophila or S. marcescens. Is this a typo?
Line 324-325: allogrooming is a poor example of a behavioral immune defense upon oral infection, as it is very unlikely to have played a role. Instead the other behavioral interactions such as trophallaxis mentioned in line 329 make much more sense. Please consider revising.
Line 348-349: I am not sure I can follow why the ubiquitous presence of multiple stresses may result in the evolution of synergies between defense mechanisms. Could you please elaborate on this?

Experimental design

no comment

Validity of the findings

Although I found the manuscript interesting there is a potential pitfall in interpreting the data. Some bacteria such as E. coli, emit a strong smell and food containing them might be avoided and only eaten at a last resort. I therefore wonder whether the continuous feeding treatment really was a continuous feeding treatment or whether a potential reluctance of the ants to feed on bacteria contaminated food might not have influenced food consumption in the Exposure treatments and thus also survival and immune gene expression. Unfortunately, the authors did not monitor food consumption. I think this is an important point and therefore the ways how this might have influenced the presented results should be thoroughly discussed.

---

## Round 0.2 · Minor Revisions

Dear Dr. Stucki and colleagues:

Thanks for re-submitting your manuscript to PeerJ. I have now received two independent reviews of your revision, and as you will see, one reviewer has raised a few more concerns about the research. Thus, I encourage you to revise your manuscript accordingly, taking into account these concerns raised by reviewer 2.

I do believe that your manuscript will be ready for publication once these issues are addressed. I look forward to seeing your revision, and thanks again for submitting your work to PeerJ.

Good luck with your revision,

-joe

Reviewer 1 ·

Basic reporting

no comment

Experimental design

no comment

Validity of the findings

no comment

Additional comments

The authors have done a great job responding to the reviewer's comments.

Reviewer 2 ·

Basic reporting

Line 274-276: replace comma with dot at the end of the sentence.

Line 331-335: I am not sure whether I agree with the logic here and I think the part on the prophylactic immune response would maybe better fit into the next paragraph. A possible formic acid ingestion would in my view have to affect bacterial survival in different ways in order to explain differences in ant host survival and gene expression.
Moreover, although the authors now included the possiblity that the ants sensed and avoided contaminated food, I miss a discussion how these might have affected their results.

Experimental design

Although I did not spot this in the previous version of the manuscript, I must agree with comment R1.24 made by the other reviewer about including a random effect of pot. As far as my understanding goes, the 20 animals in one pot are not independent and the same applies for the two pots of ingestion of bacteria (Exposure) per colony (one in each Treatment, i.e. continuous feeding or starvation). Thus, a random effect of pot would capture the interdependence of the 20 animals per pot (or the 3 animals per pot for the gene expression, but as they were pooled this is unnecessary), while the random effect of colony would capture the interdependence of the two pots per colony used for continuous feeding or starvation.
I am also curious about the way the proportional hazard test was performed on the survival model. I know this test can be performed via the function cox.zph on a coxph model call in the package survival. However, if I am running your specified model via the coxph model call with Colony as a frailty term: m<-coxph(Surv(Time1,Time2,Cens)~Treatment*Diet+frailty(Colony),data=surv) and then comparing this model with a model without the interaction: m1<-coxph(Surv(Time1,Time2,Cens)~Treatment+Diet+frailty(Colony),data=surv), I cannot get the same statistical inference you report now at the beginning of the results section for the interaction term (your report: Chi-square=7.41, df=3, p=0.06; while I get: Chi-square=7.125, df=2.9474, p=0.0654). I therefore assumed you might be using a coxme model call in the package coxme (m<-coxme(Surv(Time1,Time2,Cens)~Treatment*Diet+(1|Colony),data=surv). Doing the comparison again of models with and without the interaction gave exactly the same values as you report: Chi-square=7.4076, df=3, p=0.05998; loglik(m)=-4032.0). As far as I am aware there is currently no current equivalent to cox.zph to test proportional hazards on a coxme model. Therefore, I am wondering which model call you used (if coxme, then this should be reported in the used packages) and how you performed the proportional hazard test?
I would have also run a model including Colony and pot as a random effect but unfortunately the raw data set does not encoded a variable pot.

Validity of the findings

see comments to basic reporting and experimental design

---

## Round 0.3 · accepted · Accept

Dear Dr. Stucki and colleagues:

Thanks for re-submitting your manuscript to PeerJ, and for addressing the concerns raised by the reviewers. I now believe that your manuscript is suitable for publication. Congratulations! I look forward to seeing this work in print, and I anticipate it being an important resource for the communities studying ants and stress response. Thanks again for choosing PeerJ to publish such important work.

Good luck with your revision,

-joe

#